# Associations between indoor temperature, self-rated health and socioeconomic position in a cross-sectional study of adults in England

Joanna Sutton-Klein,[1] Alison Moody,[1] Ian Hamilton [id],[2] Jennifer S Mindell [id][1]

[1]Epidemiology and public health, UCL, London, UK
[2]UCL Energy Institute, UCL, London, UK

**Correspondence to**
Mr Ian Hamilton;
i.hamilton@ucl.ac.uk

## ABSTRACT

**Objective** Excess winter deaths are a major public health concern in England and Wales, with an average of 20 000 deaths per year since 2010. Feeling cold at home during winter is associated with reporting poor general health; cold and damp homes have greater prevalence in lower socioeconomic groups. Overheating in the summer also has adverse health consequences. This study evaluates the association between indoor temperature and general health and the extent to which this is affected by socioeconomic and household factors.

**Design** Cross-sectional study.

**Setting** England.

**Participants** Secondary data of 74 736 individuals living in England that took part in the Health Survey for England (HSE) between 2003 and 2014. The HSE is an annual household survey which uses multilevel stratification to select a new, nationally representative sample each year. The study sample comprised adults who had a nurse visit; the analytical sample was adults who had observations for indoor temperature and self-rated health.

**Results** Using both logistic and linear regression models to examine indoor temperature and health status, adjusting for socioeconomic and housing factors, the study found an association between poor health and higher indoor temperatures. Each one degree increase in indoor temperature was associated with a 1.4% (95% CI 0.5% to 2.3%) increase in the odds of poor health. After adjusting for income, education, employment type, household size and home ownership, the OR of poor health for each degree temperature rise increased by 19%, to a 1.7% (95% CI 0.7% to 2.6%) increase in odds of poor health with each degree temperature rise.

**Conclusion** People with worse self-reported health had higher indoor temperatures after adjusting for household factors. People with worse health may have chosen to maintain warmer environments or been advised to. However, other latent factors, such as housing type and energy performance could have an effect.

## Strengths and limitations of this study

► This study is based on data from 74 736 individuals, randomly sampled from the general population to be nationally representative of the population in England.
► Temperature measurements were taken using a standard protocol across the 10 years.
► Analyses used individual-level data, not ecological comparisons.
► Indoor temperatures were measured only once per participant.
► Data are cross-sectional, so causality cannot be inferred.

## INTRODUCTION

Short-term exposure to extreme hot or cold temperatures is known to be acutely harmful. For example, Analitis *et al* compared the short-term effects of cold on mortality in 15 European cities, finding that a 1°C decrease in outdoor temperature was associated with a 1.35% increase in the daily number of total natural deaths.[1] A recent systematic review has shown the adverse health consequences of high indoor temperatures, above 26°C.[2] However, the long-term effects of exposure to different temperatures are not well understood. Much of the evidence on long-term exposure to more minor deviations in ambient temperature is overwhelmingly drawn from ecological studies.[3–5]

There were 49 410 excess winter deaths (EWDs) in England and Wales over the 2017–2018 winter.[6] Although EWDs were significantly lower in 2018–2019 (23 200, provisional figure),[6] the large inequalities in EWD index between European countries suggest a major public health problem.[7] Paradoxically, while several studies have found that cooler ambient temperatures are associated with higher mortality on a day-by-day analysis, European countries with colder winters were found to have lower EWDs.[8] Studies have suggested that it is the variation in housing quality and the resulting indoor temperature differences which may instead be the drivers of these differences in health outcomes.[9 10]

Few studies have directly addressed the association between indoor temperature and health in individuals, but a number have studied known correlates of indoor temperature such as damp, insulation or perceived thermal comfort.[11][12] These studies overwhelmingly found that proxies for cool indoor temperature were associated with worse health.

Several studies have compared the health of individuals before and after housing interventions. Upgraded insulation in New Zealand was associated with increased indoor temperature, and lower levels of poor general health (OR 0.50, 95% CI 0.38 to 0.68).[13][14] Furthermore, a large pan-European study found that perceived coldness at home was strongly associated with poor general health, with an OR of 2.6 (95% CI 2.1 to 3.1),[15] a result which was replicated in a study of English adults.[16] A limitation of using perceived thermal comfort as a proxy of indoor temperature is that its subjectivity makes it vulnerable to reverse causation.

Analysis of the 1991 survey of English houses found that the strongest predictors of low indoor temperatures were age of property, absence of central heating, cost of heating, small household size (ie, fewer occupants) and low income.[17] A 2011 survey of English houses found similar importance of predictors of indoor temperature; that older properties were not the coldest; but that low energy performance ratings and low-income households had the lowest temperatures.[18]

A systematic review by Public Health England concluded that living in cold temperatures (below 18°C) may increase the risk of adverse health consequences, particularly for premature mortality in vulnerable populations.[19] However, recommendations for minimum indoor temperature are based on limited evidence.[15] Research has shown that few households are keeping the recommended indoor temperatures (ie, ≥18°C), particularly vulnerable households.[20] Recently, increasing poverty was associated with higher risk of poor housing within the UK.[21][22] Analysis of Census longitudinal study data has shown a rise in the proportion of homes with central heating from 83% in 1991, to 93% in 2001 and 98% in 2011.[23] The English Housing Survey in 2016 found that prevalence of cavity or solid wall insulation, of double glazing, and of central heating had increased in England to 49%, 83% and 92% of households, respectively.[24] A study on government interventions found that the installation of heating and insulation was effective at increasing indoor temperatures.[25] There are also deaths in England due to excessive temperatures during heatwaves,[26] so both high and low indoor temperatures warrant studying in relation to associations with health, although cold is the primary focus because that is the dominant exposure in England.[27]

The aim of our study was to investigate the relationship between indoor temperature and general health, and to examine the extent to which socioeconomic and housing variables explain any association found. We hypothesised that: socioeconomic and housing factors would be associated with indoor temperature; low indoor temperature would be associated with worse general health; and the association between temperature and general health would be stronger with decreasing socioeconomic status.

## METHODS

### Study design

This study is based on the Health Survey for England (HSE), an annual, cross-sectional, household, health examination survey which has run since 1991. A new, nationally-representative sample of the free-living general population is selected each year, using a stratified, two-stage selection process. The sampling frame is the small addresses Postal Address File. The survey varied in size each year from around 9000–16 000 households in the study sample. Data were collected first by an interviewer, who also measured height and weight; a nurse visit followed for participants who agreed. The nurse asked further questions and took biophysical measurements, including blood pressure. More details are reported elsewhere,[28][29] including the Methods report for the annual report published each year.[30]

For this survey, we used data from each year HSE 2003 to HSE 2014 excluding 2004, when the core general population sample did not have a nurse visit. These years were selected because identical variables have been coded consistently from 2003 onwards; 2014 was the most recent dataset available in the UK Data Archive at the time the analyses commenced.

### Participants

The study sample was adults aged 16 years and over who took part in the nurse visit component of the HSE in 2003 or 2005 to 2014. The study sample comprised adults who had a nurse visit; the analytical sample was adults who had observations for indoor temperature and self-rated health (online supplemental figure S1). The analytical sample (n=74 736) was compared with those excluded from the study sample (n=2065, table 1 and online supplemental table S1). There were no statistically significant differences between the analytical sample and those excluded regarding mean monthly outdoor temperatures by region (data not shown). However, all categorical variables, except for respiratory disease, had significant differences between the analytical sample and the excluded participants. The analytical sample comprised 55% women, while 74% of the excluded participants were women (p<0.01). The excluded survey participants had a younger age profile than the analytical sample; were less likely to be in the top two income quintiles; were more likely to live in social housing; but were more likely to have degree level qualifications. The analytical sample were more likely to own their homes, and to have a longstanding illness or cardiovascular condition.

Participants gave informed consent at each stage of the survey.

**Table 1** Distribution of categorical variables by complete and incomplete cases within the study sample, and p values for $\chi^2$ tests (unweighted)

| Variable | | Incomplete (n=2065) | | Complete (n=74 736) | | $\chi^2$ p value |
|---|---|---|---|---|---|---|
| | | N | % | N | % | |
| Sex | Men | 533 | 26 | 33 526 | 45 | <0.01 |
| | Women | 1532 | 74 | 41 210 | 55 | |
| Age group | 16–24 | 373 | 18 | 6539 | 9 | <0.01 |
| | 25–34 | 639 | 31 | 9586 | 13 | |
| | 35–44 | 375 | 18 | 13 145 | 18 | |
| | 45–54 | 179 | 9 | 12 929 | 17 | |
| | 55–64 | 161 | 8 | 12 753 | 17 | |
| | 65–74 | 162 | 8 | 11 315 | 15 | |
| | 75+ | 176 | 9 | 8412 | 11 | |
| NS-SEC5 | Managerial/professional | 557 | 27 | 25 297 | 34 | <0.01 |
| | Intermediate occupation | 234 | 11 | 10 354 | 14 | |
| | Small employers | 9 | 0 | 6514 | 9 | |
| | Lower supervisory | 96 | 5 | 6087 | 8 | |
| | Semiroutine | 511 | 25 | 23 314 | 31 | |
| | Missing | 658 | 32 | 3170 | 4 | |
| Education | Degree level or higher | 476 | 24 | 15 400 | 21 | <0.01 |
| | Qualification below degree | 992 | 50 | 40 233 | 54 | |
| | No qualification | 515 | 26 | 19 103 | 26 | |
| Income quintile | Lowest quintile | 420 | 20 | 10 861 | 15 | <0.01 |
| | Second lowest | 302 | 15 | 12 115 | 16 | |
| | Third lowest | 260 | 13 | 13 190 | 18 | |
| | Fourth lowest | 265 | 13 | 13 537 | 18 | |
| | Highest quintile | 321 | 16 | 13 291 | 18 | |
| | Missing | 497 | 24 | 11 742 | 16 | |
| Tenure | Owner-occupied | 1076 | 52 | 55 159 | 74 | <0.01 |
| | Privately rented | 283 | 14 | 6476 | 9 | |
| | Socially rented | 492 | 24 | 11 347 | 15 | |
| | Missing | *214* | *10* | *1174* | *2* | |
| Household size | 1 | 284 | 14 | 13 677 | 18 | <0.01 |
| | 2 | 712 | 35 | 30 117 | 40 | |
| | 3 | 540 | 26 | 12 585 | 17 | |
| | 4 | 323 | 16 | 12 173 | 16 | |
| | 5+ | 206 | 10 | 6184 | 8 | |
| General health | Fair | 1555 | 76 | 55 235 | 74 | 0.06 |
| | Poor | 498 | 24 | 19 501 | 26 | |
| | Missing | 12 | 1 | 0 | 0 | |
| Longstanding illness | No | 1269 | 62 | 39 246 | 53 | <0.01 |
| | Yes | 781 | 38 | 35 490 | 48 | |
| | Missing | 15 | 1 | 0 | 0 | |
| Heart condition | No | 1838 | 90 | 63 905 | 86 | <0.01 |
| | Yes | 212 | 10 | 10 831 | 14 | |
| | Missing | *15* | *1* | *0* | *0* | |

Continued

| Variable | | Incomplete (n=2065) | | Complete (n=74 736) | | $\chi^2$ p value |
|---|---|---|---|---|---|---|
| | | N | % | N | % | |
| Respiratory condition | No | 1869 | 91 | 68 051 | 91 | 0.86 |
| | Yes | 181 | 9 | 6685 | 9 | |
| | Missing | 15 | 1 | 0 | 0 | |
| Total | | 2065 | 3 | 74 736 | 97 | |

NS-SEC5, National Statistics-Socio-Economic Classification of occupation in five categories.

## Data

### General health

General health was a self-assessed variable, referred to as self-rated health. It was recorded at the interview stage, prior to the nurse visit. The participants selected one of the five answer options "very good, good, fair, bad or very bad?" to the question: "How is your health in general?" The wording of this question, which has been used throughout the HSE series, is identical to that recommended by the WHO in its 1996 report on harmonisation of instruments in health interview surveys[31]; the the Office for National Statistics (ONS) in 2008[32]; and the UK Government Statistical Service harmonisation.[32] For this analysis, general health was dichotomised into 'good', which encompassed the categories very good and good, and 'poor', which included fair, bad and very bad. Fair was placed in the 'poor' category rather than the 'good' category as the distribution of general health is highly skewed: 74% of participants reported good or very good general health.

### Poor health

The other self-reported health data used in these analyses included the presence of limiting or non-limiting long-standing illness; a respiratory condition or a cardiovascular condition. Participants were first asked if they had any longstanding illness or condition. Those who responded positively were then asked if it limited their daily activities, and asked what the condition(s) were, to a maximum of six. These were then coded into ICD-10 chapters (International Statistical Classification of Diseases and Related Health Problems (ICD) 10th Version (ICD-10)), from which responses from participants volunteering a respiratory condition (chapter J) and/or a cardiovascular condition (chapter I) were identified.

### Demographic data

Gender and age were collected at the interview.

### Socioeconomic data

Three variables were used for socioeconomic position. Education level was categorised as degree or equivalent, qualification below degree or no qualification. Equivalised household income, which adjusts for the numbers of adults and of children in the household, was divided into quintiles. The National Statistics-Socio-Economic Classification of occupation in five categories (NS-SEC5)[33] was derived from detailed information about employment status, occupation, size of organisation and managerial responsibility. The categories were managerial and professional, intermediate, small employers, lower supervisory and technical and semiroutine.

### Housing

Housing information comprised the number of occupants in the household, and housing tenure. The latter was categorised as owner occupied (eg, owned outright or with a mortgage), rented privately or socially rented (ie, from a housing association or local government).

### Temperature

Indoor air temperature was measured by the nurse, using a digital thermometer with a probe, prior to measuring blood pressure. The thermometer was placed on a surface near the blood pressure equipment, away from a radiator and out of direct sunlight. Nurses were recommended that the probe hang over the edge of the table.

Mean monthly outdoor temperature data by region were obtained from the Met Office website. The data are publicly available. The Met Office is a governmental agency in the UK which provides weather forecasts and warnings. The data were measured according to their standard protocol at weather stations 40 kilometres apart across the country, every hour.

### Data sharing

The pseudonymised HSE data from each survey are deposited in the UK Data Archive and are available to be downloaded from the UK Data Service at: https://beta.ukdataservice.ac.uk/datacatalogue/series/series?id=2000021.

### Statistical analysis

For the variables NS-SEC5, income and tenure, answers in the 'other' category or categories outside of the main analytical categories (eg, unemployed, student or homemaker for NS-SEC) were grouped into an additional category of 'other or missing' and included in the models. This was because there were many participants in those categories and excluding them would have substantially

decreased the sample size. The data were combined into a single dataset.

The analysis was run in Stata V.15 (StataCorp). Models were built to test each hypothesis. Each model used the same analytical sample. The models were built using maximum likelihood estimation, and the best models were selected using the likelihood ratio test. Age was grouped for analysis as 16–24 then 10 years groups to 75+, to avoid prior assumptions of the shape of the relationship.

Associations between socioeconomic and housing variables and indoor temperature were explored in two ways. Logistic regression models were created with indoor temperature <18°C as the outcome variable (compared with indoor temperature of 18°C or above) as well as linear regression models with indoor temperature as a continuous variable. For both sets of models, socioeconomic and housing variables were included as explanatory variables. Missing data among the covariates were included in the regression as an additional category.

To test the association of indoor temperature with general health, a similar set of logistic regression models were created but with indoor temperature as the explanatory variable, and general self-reported health as the outcome variable, again adjusting for demographic, housing and socioeconomic variables. The models were run for the whole sample, then were then repeated stratified by tertile of outdoor temperature to assess if the association differed across the range of outdoor temperatures. As a sensitivity analysis, model 9 (fully adjusted) was repeated stratified by tertile of indoor temperature to assess if the association differed across the range of indoor temperatures.

The unadjusted coefficients of the variables are reported, as well as the coefficients after adjusting for other socioeconomic and housing variables. Collinearity was assessed by looking for reversal of the sign of the coefficients and the variance inflation factor.

The analysis was weighted using the survey-provided weights, which account for probabilities of selection (due to the sample design), and response rate differences by age, sex and region. This ensured a weighted sample which matched the ONS-mid-year population estimates by age and sex for each survey year.[30] The multistage survey design was accounted for using the 'svy' commands in STATA, using the survey's primary sampling units, and using government office region as the strata variable, since this was a component of stratification that was consistent across the survey years.

## RESULTS
The analytical sample consisted of 74 736 participants, 97.3% of the study sample. Their characteristics are shown in table 1 and online supplemental table S1. The indoor temperature ranged from 7.5°C to 36.8°C, with a mean of 20.7°C (SD 2.3). The outdoor temperature ranged from −1.2°C to 20.5°C, with a mean of 10·1°C (SD 4·6). For both indoor and outdoor temperature, most

of the variation was attributable to month. Ten per cent of households had indoor temperatures below the UK government's recommended minimum of 18°C.

### Socioeconomic and housing variables and indoor temperature
Households with more than one occupant had a lower odds of an indoor temperature <18°C (table 2) and higher mean indoor temperatures (table 3 and online supplemental table S1) than single person households, as did households with higher equivalised household incomes compared with those with lower incomes. The adjusted OR (table 2) and β-coefficient (table 3) for both household size and income showed a gradient across all categories. Higher educational attainment, and private sector rented houses (compared with owner-occupied housing) were associated with lower mean indoor temperatures (table 3) and a higher odds of the indoor temperature being <18°C (table 2). NS-SEC5 was variably associated with low temperature: compared with those in managerial and professional occupations, we found reduced adjusted OR among those in intermediate occupations but higher odds among small employers, and with mean indoor temperature (positive association in lower supervisory and technical households but inverse association among small employers) in the fully adjusted models.

### Indoor temperature and health
Each one degree increase in indoor temperature was associated with a 1.4% (95% CI 0.5% to 2.3%) increase in odds of poor health, after adjusting for age and gender (see figure 1). After adjusting for socioeconomic and housing factors, each degree increase in temperature is associated with a 1.7% (95% CI 0.7% to 2.6%) increase in odds of poor health.

A series of regression models were run to examine the impact on the odds of reporting poor health associated with indoor temperature of demographic, housing and socioeconomic factors (table 4). Adjusting for household size increased the OR most, with small changes for household size and tenure or for equivalised household income. Adjusting for tenure, NS-SEC5, or all three socioeconomic factors reduced the OR a little. The linear association between indoor temperature and reporting poor health remained significant in each model; the final model, including all these covariates, also increased the OR compared with the base model.

When stratified by indoor temperature range, and adjusting for socioeconomic and housing factors, the ORs for each indoor temperature tertile did not differ significantly from one another (sensitivity analysis, data not shown). With the truncated range of temperatures, the relationship between indoor temperature and poor health was no longer significant within the upper two tertiles. This may be due to truncation in the distribution of indoor temperature, meaning that only the lowest tertile included the low temperatures associated with poor health.

**Table 2** Logistic regression models of associations of socioeconomic and housing variables with low indoor temperature (<18°C)

| | Minimally adjusted* | | | Fully adjusted† | | |
|---|---|---|---|---|---|---|
| | OR | 95% CI | P value | aOR | 95% CI | P value |
| **Household size** | | | | | | |
| 1 | 1 | | | 1 | | |
| 2 | 0.58 | 0.54 to 0.63 | <0.001 | 0.59 | 0.54 to 0.63 | <0.001 |
| 3 | 0.46 | 0.41 to 0.51 | <0.001 | 0.46 | 0.41 to 0.51 | <0.001 |
| 4 | 0.38 | 0.34 to 0.43 | <0.001 | 0.38 | 0.34 to 0.43 | <0.001 |
| 5+ | 0.37 | 0.32 to 0.44 | <0.001 | 0.36 | 0.31 to 0.43 | <0.001 |
| **Tenure** | | | | | | |
| Owner occupied | 1 | | | 1 | | |
| Privately rented | 1.49 | 1.33 to 1.68 | <0.001 | 1.27 | 1.12 to 1.43 | <0.001 |
| Socially rented | 1.10 | 1.00 to 1.20 | 0.056 | 0.94 | 0.85 to 1.05 | 0.264 |
| Other or missing | 1.39 | 1.15 to 1.68 | 0.001 | 1.18 | 0.98 to 1.43 | 0.086 |
| NS-SEC5 | | | | | | |
| Managerial and professional | 1 | | | 1 | | |
| Intermediate | 0.90 | 0.83 to 0.98 | 0.015 | 0.92 | 0.84 to 1.00 | 0.047 |
| Small employers | 1.17 | 1.06 to 1.29 | 0.003 | 1.20 | 1.09 to 1.33 | <0.001 |
| Lower supervisory and technical | 0.91 | 0.81 to 1.01 | 0.083 | 0.93 | 0.83 to 1.05 | 0.236 |
| Semiroutine | 1.00 | 0.94 to 1.08 | 0.901 | 1.02 | 0.94 to 1.11 | 0.585 |
| Other or missing | 1 | 0.85 to 1.18 | 0.98 | 1.12 | 0.95 to 1.32 | 0.165 |
| **Education** | | | | | | |
| Degree | 1 | | | 1 | | |
| Qualification below degree | 0.91 | 0.85 to 0.98 | 0.017 | 0.92 | 0.85 to 1.00 | 0.045 |
| No qualification | 0.90 | 0.82 to 0.99 | 0.029 | 0.84 | 0.75 to 0.93 | 0.001 |
| **Equivalised household income quintile** | | | | | | |
| Lowest quintile | 1 | | | 1 | | |
| 2nd quintile | 0.87 | 0.78 to 0.98 | 0.024 | 0.91 | 0.81 to 1.03 | 0.126 |
| 3rd quintile | 0.86 | 0.77 to 0.96 | 0.010 | 0.88 | 0.78 to 0.99 | 0.030 |
| 4th quintile | 0.84 | 0.75 to 0.95 | 0.004 | 0.83 | 0.73 to 0.94 | 0.003 |
| Highest quintile | 0.86 | 0.77 to 0.97 | 0.013 | 0.77 | 0.67 to 0.88 | <0.001 |
| Other or missing | 0.87 | 0.77 to 0.99 | 0.035 | 0.9 | 0.79 to 1.03 | 0.112 |

*Minimally adjusted models: adjusted for outdoor temperature and age group.
†Fully adjusted models: mutually adjusted for all socioeconomic and housing variables in this table.
aOR, adjusted OR; NS-SEC5, National Statistics-Socio-Economic Classification of occupation in five categories.

When regression models were repeated stratified for tertile of outdoor temperature, we found the ORs of indoor temperature on poor health were stronger for the coldest tertile (ie, <7°C). For example, the adjusted ORs for models 7 and 9, which included adjustment for housing variables, were 1.04 (1.02–1.05) in the lowest tertile but were not significantly different from 1.0 for the middle and highest tertile of outdoor temperature (table 4). This may be because the lower the outdoor temperature, the more likely it is that the occupant will choose to turn on their heating and therefore choose what temperature their home is at, which will depend on the characteristics of both the home and the residents.

On warmer days, occupants are less likely to turn on their heating, and therefore, the indoor temperature will be determined more by the outdoor temperature than the characteristics of the occupant, and as such will be less associated with any characteristics of the occupant.

## DISCUSSION

The results showed an association between higher indoor temperatures and increased odds of poor self-rated health. This has also been shown recently in a small number of individuals in Cornwall, UK: participants with poorer health maintained their homes

**Table 3** Linear regression models with socioeconomic and housing variables as explanatory variables for indoor temperature

| | Minimally adjusted* | | | Fully adjusted† | | |
|---|---|---|---|---|---|---|
| | β | 95% CI | P value | β | 95% CI | P value |
| **Household size** | | | | | | |
| 1 | Ref | | | | | |
| 2 | 0.41 | 0.35 to 0.46 | <0.001 | 0.43 | 0.38 to 0.49 | <0.001 |
| 3 | 0.63 | 0.56 to 0.71 | <0.001 | 0.65 | 0.58 to 0.73 | <0.001 |
| 4 | 0.74 | 0.66 to 0.82 | <0.001 | 0.76 | 0.68 to 0.85 | <0.001 |
| 5+ | 0.79 | 0.68 to 0.90 | <0.001 | 0.81 | 0.70 to 0.93 | <0.001 |
| **Tenure** | | | | | | |
| Owner occupied | Ref | | | | | |
| Privately rented | −0.23 | −0.32 to −0.14 | <0.001 | −0.11 | −0.20 to −0.02 | 0.017 |
| Socially rented | 0.13 | 0.06 to 0.19 | <0.001 | 0.21 | 0.14 to 0.28 | <0.001 |
| Other or missing | −0.12 | −0.028 to 0.04 | 0.146 | −0.01 | −0.17 to 0.15 | 0.918 |
| NS-SEC5 | | | | | | |
| Managerial and professional occupations | Ref | | | | | |
| Intermediate | 0.08 | 0.03 to 0.13 | 0.003 | 0.03 | −0.03 to 0.08 | 0.298 |
| Small employers | −0.05 | −0.12 to 0.02 | 0.177 | −0.11 | −0.18 to −0.04 | 0.001 |
| Lower supervisory and technical | 0.18 | 0.11 to 0.24 | <0.001 | 0.09 | 0.02 to 0.16 | 0.010 |
| Semi-routine | 0.06 | 0.01 to 0.11 | 0.020 | −0.04 | −0.09 to 0.01 | 0.128 |
| Other or missing | 0.11 | 0.01 to 0.22 | 0.040 | −0.06 | −0.17 to 0.04 | 0.250 |
| **Education** | | | | | | |
| Degree | Ref | | | | | |
| Qualification below degree | 0.16 | 0.11 to 0.21 | <0.001 | 0.14 | 0.08 to 0.19 | <0.001 |
| No qualification | 0.22 | 0.16 to 0.29 | <0.001 | 0.22 | 0.15 to 0.29 | <0.001 |
| **Income quintile** | | | | | | |
| Lowest quintile | Ref | | | | | |
| Second quintile | 0.03 | −0.05 to 0.11 | 0.468 | 0.04 | −0.05 to 0.12 | 0.402 |
| Third quintile | 0.02 | −0.06 to 0.10 | 0.631 | 0.07 | −0.01 to 0.16 | 0.081 |
| Fourth quintile | 0.01 | −0.07 to 0.09 | 0.843 | 0.11 | 0.02 to 0.19 | 0.016 |
| Highest quintile | −0.04 | −0.12 to 0.05 | 0.402 | 0.15 | 0.06 to 0.24 | 0.001 |
| Other or missing | 0.05 | −0.04 to 0.14 | 0.315 | 0.08 | −0.01 to 0.17 | 0.091 |

*Minimally adjusted models: adjusted for outdoor temperature and age group.
†Fully adjusted models: mutually adjusted for all socioeconomic and housing variables in this table.
NS-SEC5, National Statistics-Socio-Economic Classification of occupation in five categories.

at higher temperatures.[34] However, the relationship between socioeconomic and housing factors with indoor temperature was complicated, with different measures having different directions of association with indoor temperature (figure 2). The most intuitive association is that those with lower incomes tended to have lower indoor temperatures, possibly because they cannot afford to pay for gas or electricity to heat their homes. This may explain why the association between indoor temperature and poor health was strongest when the outdoor temperature was lower. Conversely, other markers of lower social class, including lower NS-SEC5 and lower education, were associated with warmer indoor temperatures. These differing associations might

be best explained by the finding that privately rented accommodation was cooler than homes that were socially rented. This is a reflection that the ability to raise indoor temperature is determined by both the financial ability to pay for fuel and the insulation of the home. Socially rented accommodation tends to be better insulated than privately rented accommodation both because of the housing quality and space standards required on social housing[35] and because social landlords such as housing associations and local councils may have more incentive to provide decent quality accommodation as a long-term asset. A small study in Japan, using perceived thermal comfort not measured temperature, found that people who reported living in cold homes had a higher risk of

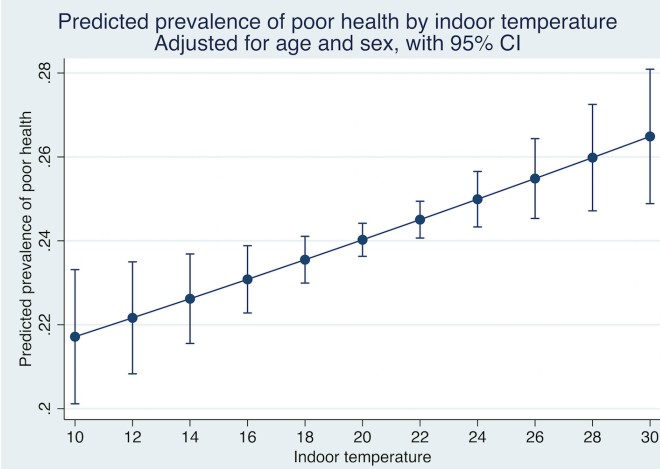

**Figure 1** Predicted prevalence of poor health for each degree increase in indoor temperature. Prevalence of poor health predicted by the logistic regression of indoor temperature on poor health, adjusted for age and sex. OR 1.01 (95% CI 1.00 to 1.02), see table 4, model 1.

frailty—but only among those who were dissatisfied with their economic situation.[36]

It is not clear from this research why temperatures are higher among households with worse self-rated health and whether this is due to personal preferences and behaviours; advice from medical or other professionals to maintain a warmer home; the difference in the type of building they occupy (ie, dwellings that are typically warmer such as smaller houses or flats, and social housing rather than privately rented); or some combination of these. Figure 2 shows a conceptual framework for the way in which these variables affect the associations.

The strengths of this study are the large number of participants, who were nationally representative of the non-institutionalised population living in England. The temperature measurements were taken using a standard protocol across the time period but were not the focus of the surveys, so the likelihood of differential response rates systematically associated with indoor temperature is very small. Unlike the study by Armstrong and colleagues, we report data based on individuals, rather than ecological analyses.[37]

Limitations include the significant differences in many characteristics between the analytical sample, who had indoor measurements, and those excluded because there were no measurements. The latter comprises those who were ineligible for blood pressure measurement (eg, being pregnant) or declined the measurement. However, fewer than 3% of the study sample were excluded; the analytical sample is, therefore, almost identical to the study sample overall. In addition, there were some missing data among the analytical sample (table 1). Income is the variable with the highest level of missingness. It is likely that this group contains disproportionate numbers of households both of high and of low income. Tables 2 and 3 show that this group have a similar odds of low temperature as the second lowest quintile, and a similar coefficient for linear

**Table 4** Logistic regression models showing ORs of poor health for indoor temperature, stratified by tertile of outdoor temperature

| Indoor temperature*† | All | | | Lowest tertile of outdoor temperature (<7°C) | | | Middle tertile of outdoor temperature (7°C–12.8°C) | | | Highest tertile of outdoor temperature (≥12.9°C) | | |
|---|---|---|---|---|---|---|---|---|---|---|---|---|
| | aOR | 95% CI | P value | aOR | 95% CI | P value | aOR | 95% CI | P value | aOR | 95% CI | P value |
| Model 1: adjusted for age and gender | 1.01 | 1.00 to 1.02 | 0.003 | 1.02 | 1.01 to 1.04 | 0.004 | 1.01 | 1.00 to 1.03 | 0.122 | 1.02 | 1.00 to 1.03 | 0.069 |
| Model 2: model 1+household size | 1.02 | 1.01 to 1.03 | <0.001 | 1.04 | 1.02 to 1.05 | <0.001 | 1.02 | 1.00 to 1.04 | 0.018 | 1.02 | 1.00 to 1.03 | 0.032 |
| Model 3: model 1+tenure | 1.01 | 1.00 to 1.02 | 0.014 | 1.03 | 1.01 to 1.05 | 0.001 | 1.01 | 0.99 to 1.03 | 0.275 | 1.00 | 0.99 to 1.02 | 0.614 |
| Model 4: model 1+NS-SEC5 | 1.01 | 1.00 to 1.02 | <0.005 | 1.02 | 1.01 to 1.04 | 0.007 | 1.02 | 1.00 to 1.03 | 0.091 | 1.01 | 1.00 to 1.03 | 0.163 |
| Model 5: model 1+income | 1.02 | 1.01 to 1.03 | <0.001 | 1.03 | 1.01 to 1.05 | 0.001 | 1.01 | 1.00 to 1.03 | 0.138 | 1.01 | 1.00 to 1.03 | 0.117 |
| Model 6: model 1+education | 1.01 | 1.00 to 1.02 | 0.034 | 1.02 | 1.00 to 1.03 | 0.029 | 1.01 | 0.99 to 1.03 | 0.276 | 1.01 | 0.99 to 1.03 | 0.208 |
| Model 7: model 1+housing variables | 1.02 | 1.01 to 1.02 | 0.002 | 1.04 | 1.02 to 1.05 | <0.001 | 1.01 | 1.00 to 1.03 | 0.132 | 1.01 | 0.99 to 1.02 | 0.470 |
| Model 8, model 1+socioeconomic variables | 1.01 | 1.00 to 1.02 | 0.007 | 1.02 | 1.01 to 1.04 | 0.004 | 1.01 | 0.99 to 1.03 | 0.191 | 1.01 | 0.99 to 1.03 | 0.310 |
| Model 9: model 1+housing and socioeconomic variables | 1.02 | 1.01 to 1.03 | <0.001 | 1.04 | 1.02 to 1.05 | <0.001 | 1.02 | 1.00 to 1.04 | 0.086 | 1.01 | 0.99 to 1.02 | 0.465 |

*Indoor temperature was linear in these models.
†Models are listed in order of increasing likelihood. All models adjusted for age group and gender; model 7 as model 1 plus household size and tenure; model 8 as model 1 plus NS-SEC5, equivalised income quintiles, and education; model 9 included all the variables.
aOR, adjusted OR; NS-SEC5, National Statistics-Socio-Economic Classification of occupation in five categories.

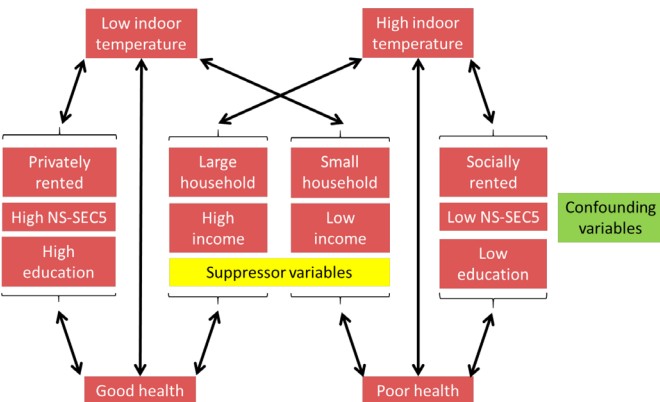

**Figure 2** Conceptual framework for temperature-health association. NS-SEC5, National Statistics-Socio-Economic Classification of occupation in five categories.

temperature as the third quintile in the fully adjusted models.

The third main limitation was the measurement of indoor temperature at a single time point for each participant in whichever room they were 'hosting' the survey nurse, generally in the living room or kitchen. Huebner *et al* have shown four different profiles in living room temperature variability across the day, varying from bimodal (morning and evening peaks), steadily increasing (shallow and steep) to flat with little variation.[38] Using the single time point temperature reading in the analysis requires an assumption that the snapshot temperature reading for each participant reflects an average temperature for the home over a significant period of time.

Hajat and Gasparrini have shown that 40% of cold-related deaths occur outside the December–March period used for calculating 'EWDs'.[27] For that reason, and given seasonal and temperature differences across the regions of England, we used tertile of outdoor temperature rather than season of other groupings of months. We showed a strong relationship at low outdoor temperatures but little if any effect at high outdoor temperatures. Impacts of heatwaves are generally acute, so it is less likely that we would have found a strong relationship at higher outdoor temperatures.

While the subjectivity of self-rated health makes it vulnerable to information bias, it has been shown to be highly predictive of future mortality and health outcomes, and is used frequently across the scientific literature.[39 40] Self-rated health is strongly associated with total mortality[41–45] and with some specific causes of death[46 47] but not with external causes.[46] It also predicts functional ability.[48] Self-rated health is explained more by measures of mental and physical health than by demographic or socioeconomic factors.[49]

Apart from the well-known association between temperature and blood pressure,[50–53] there is less evidence pointing at more specific health outcomes related to temperature. Thus, as this is the first study of its kind, the broadness of general health as the outcome was a strength rather than a weakness. The

differences between the entire sample and those with complete data on self-rated health and temperature was primarily due to non-response to the nurse visit among interviewees. This was dealt with by use of non-response weighting.[29]

While guidelines on healthy indoor temperatures have been issued by WHO and the UK Public Health England, they appear arbitrary and the WHO itself describe its own temperature recommendations as '(scientifically) weak and not well known to the public or policy-makers'.[54–56] Since 2014, the recommendation from the UK government is for a minimum of 18°C in all rooms, a change from the previous 21°C for living rooms.[57] The government attributed the change to a review of evidence, awareness of climate change and financial difficulties, and acceptability ('We know that people have strong feelings about their homes and don't want to be told what to do in them').[56] Further research is needed to understand the temporality and mechanism of the association.

A recent study in England found no impact of Winter Fuel Payment eligibility on indoor temperature,[58] but no study has examined at individual level the impacts of energy efficiency retrofits on indoor temperature, residents' heating-related behaviours, or health. When linked data are available, we intend to examine the associations of indoor temperature with subsequent hospital admissions and mortality. This will be important for assessing and making health-related policy recommendations for indoor temperature. Future research in this area could take advantage of internet-enabled 'smart' thermostats.

**Correction notice** This article was previously published with wrong licence. The correct licence for the paper is CC-BY.

**Acknowledgements** We thank the nurses and participants in the Health Surveys for England, and our colleagues at NatCen Social Research and UCL.

**Contributors** This work was conducted as part of JS-K's MSc in Social Epidemiology. She led on conducting the literature search, designing and conducting the analyses, creating the figures, and interpreting the data. AM and JSM contributed advice to the dissertation. JS-K and JSM wrote the initial draft of the manuscript. IH updated the literature search and contributed to analysis and preparation of the manuscript. All authors contributed to subsequent drafts, approved the final and revised manuscripts, and are accountable for the accuracy and integrity of all aspects of the work.

**Funding** NHS Digital, which funds the HSE, also funds AM and JSM to work on the annual HSE. This work was also supported by the EPSRC (EP/R035288/10 to fund IH to work on home energy efficiency.

**Disclaimer** NHS Digital and EPSRC played no part in the planning, conduct, or interpretation of these analyses; the drafting of the manuscript; nor the decision to publish.

**Competing interests** JSM and AM report grants from NHS Digital during the conduct of the study; IH report grants from the Engineering and Physical Sciences Research Council (EPSRC).

**Patient consent for publication** Not required.

**Ethics approval** Research ethics approval was obtained prior to each year's survey from the relevant research ethics committees. Ethical approval was not required for this specific study, which was conducted on non-identifiable, archived data held by the UK Data Service for widespread research use.

**ORCID iDs**

Ian Hamilton http://orcid.org/0000-0003-2582-2361
Jennifer S Mindell http://orcid.org/0000-0002-7604-6131

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
