## [Reviewer comments · BMJ Open]

ARTICLE DETAILS

TITLE (PROVISIONAL)	Associations between indoor temperature, self-rated health and socio-economic position in a cross-sectional study of adults in England
AUTHORS	Mindell, Jennifer; Sutton-Klein, Joanna; Moody, Alison; Hamilton, Ian

VERSION 1 – REVIEW

REVIEWER	Keigo Saeki Nara Medical University School of Medicine, Nara, Japan
REVIEW RETURNED	19-May-2020

GENERAL COMMENTS	General comment: The author investigates the cross-sectional association between indoor temperature and self-rated health based on large sample data from a national survey in England, and they reported a significant association between higher indoor temperature and poor self-rated health independent of age, gender, and socioeconomic status. The findings are important to develop a suitable housing environment for disease prevention, but there are some issues to be clarified. Major comment 1 The regression line in figure1 shows a linear association between higher indoor temperature and a higher probability of poor health. However, we should pay attention to the consistency of the association in the range of low, middle, and high indoor temperatures. The author stated that the model was stratified by indoor temperature to assess if the direction of the association differed across the range of indoor temperatures (line 55 page5), but please indicate the present a stratified analysis by three or four categories of indoor temperatures. Major comment 2 The author should consider the confounding effect of outdoor temperature in the association between indoor temperature and self-rated health. Please conduct multivariable models adjusting for outdoor temperature in Table 3 and Table 4. In addition, stratified analysis by four seasons will be informative for readers. The association between indoor temperature and self-rated health independent of outdoor temperature will suggest valuable information to readers to make a prevention measure. Major comment 3 We can not find information about the basic characteristics of participants such as age, gender, BMI, household size, Tenure, NS-SEC5, Education, and household income, indoor temperature, and outdoor temperature. Please present the summary data (mean or
--

	median of continuous data, and proportion and number of categorical data) of these variables by the seasons of measurement as regular Table. The author states that their characteristics are shown in supplementary Table S1 (page 7, line 12). But I can not find Table S1 in the proof file and uploaded the supplementary file. Minor comment 1 If possible, please present the validity of the questionnaire of self-rated health. Minor comment 2 Please explain the method to calculate the “% change in OR” in Table 3 in the method section. Minor comment 3 Please explain the statistical method of figure 1, and please indicate the parameters of the linear regression, such as regression coefficient, correlation coefficient, and p-value in text. The term “probability” of poor health should be replaced by “Prevalence” (figure 1). Minor comment 4 Please add a detailed explanation about the potential mechanism in the discussion section using figure 2.
--	---

REVIEWER	Steven M. Schmidt Lund University Sweden
REVIEW RETURNED	07-Jul-2020

GENERAL COMMENTS	In general, I think this is an important topic, and it could be a useful paper. The large sample is rather unique. However, a one time temperature reading may not be very indicative of what the normal indoor temperature is in the areas where the resident spends their time. As most of the previous research in this area has been with older adults, the large number of younger adults could account for some of the findings. Younger adults, for one, spend much less time at home than older adults, so the indoor temperature at home has a smaller impact on their health compared to older adults. There are a number of issues addressed below that would need to be resolved as some parts of the paper are very difficult to understand in the present state, so it is difficult to fully judge the findings at this time. The main focus seems to be on cold homes in the abstract objective, but you do assess a full range of temperatures in at least one analysis. The focus should be more clear in the background as well. Please be consistent with terminology throughout. For example in the abstract you talk about ill-health, but then use poor health in the body of the paper. You also talk about “Ill households” in the abstract but a household cannot be ill, only a person can. In general the abstract uses different language and terms compared to the body of the paper. Research Hypotheses The second hypothesis is “poor indoor temperature control would be associated with worse general health.” However, you did not assess temperature control. It is not clear what you mean by poor indoor temperature control?
---

	Methods I would like to see a description of the study design at the start of the methods. In the title and abstract you state that it is a “cross-sectional study” but this is not stated in the methods or elsewhere in the body of the paper. A little more detail of the design would also be nice. Participants The “Participants section has a lot of detail that is not per se related to the description of participants. Most of the first paragraph in the participants section is used to describe the HSE survey. I would suggest splitting this into two separate sections with one describing your actual sample and the other describing the survey and data collection procedures. When the survey is described it would also be good to give a short description of how the data were combined. Where all people from all years combined into a single data set? Was any adjustment made for the year of assessment? Over an 11 year period there could be other factors that could have influenced perceived health. Data The second paragraph is hard to follow and seems to have some incorrect use of punctuation. Please check for proper use of colon and semi colon in first sentence. I would recommend re-writing this second paragraph of Data to clarify what was assessed and what the response choices were for each item. I had to look down to table 1 to really start to understand what the variables were. This should already be clearly explained in the methods. The response choice need clear explanations for an international readership to understand. It seems very UK specific, so I have trouble knowing what they mean unless defined. For example in the three categories for education, what type of degree (university, primary school, secondary school)? What is and “other qualification”? Statistical analysis I am not sure I understand your justification for categorizing age. Please elaborate to explain this. If you are going to include a category of participants as “other or missing”, then I think they should also be shown in the results. The analyses are not described for the results presented in Table 1. Please add it. It seems that the temperature was dichotomize into <18 degrees and 18 degrees or higher, but this is not stated. In general it is hard to match what is described in this section with the results. Please elaborate and clarify what was done. Results There is no “Supplementary Table S1” that I could find. There needs to be a very good description of the study sample, but I cannot judge this at this point as it seems like it might be in the missing table. In the first paragraph of results indoor temperature is presented but outdoor temperature is not. Please show the range and mean for outdoor temperature as well. Please organize the paper so that the methods (Statistical analyses) and results are the same order as the hypotheses. It is currently not consistent There seems to be an extra figure with no explanation. The last row of Table 2 has some numbers in bold, but I am not sure why. In the section “Indoor temperature and health” you describe the increase in odds, but then you refer to figure 1 where you label the axis probabilities. What is really being presented in the two parts?
--	--

	Probabilities are not mentioned in any of the text and only in the figure. Several terms are used in the results: odds, odds ratios, and probability, but it is not clear if you have calculated different things or if these terms are, incorrectly, being used interchangeably. Discussion I would like to see more discussion about what the finding mean. You simply state that “the relationship between socioeconomic and housing factors with indoor temperature was complicated”, but you do not really make any effort to try to explain that complexity. Besides mortality, a number of studies have looked at the impact of temperature on blood pressure, and have shown larger increases in blood pressure with lower temperatures and higher ages. Perhaps the indoor temperature does not have much of an association among younger ages, and you may find a different relationship at higher ages. As there was a nurse visit what types of health outcomes were assessed that you could use as an alternative to self-reported health? I would say blood pressure for one.
--	--

VERSION 1 – AUTHOR RESPONSE

Reviewer: 1

Reviewer Name: Keigo Saeki

Institution and Country: Nara Medical University School of Medicine, Nara, Japan

Please state any competing interests or state 'None declared': none

General comment:

The author investigates the cross-sectional association between indoor temperature and self-rated health based on large sample data from a national survey in England, and they reported a significant association between higher indoor temperature and poor self-rated health independent of age, gender, and socioeconomic status. The findings are important to develop a suitable housing environment for disease prevention, but there are some issues to be clarified.

Thank you

Major comment 1: The regression line in figure1 shows a linear association between higher indoor temperature and a higher probability of poor health. However, we should pay attention to the consistency of the association in the range of low, middle, and high indoor temperatures. The author stated that the model was stratified by indoor temperature to assess if the direction of the association differed across the range of indoor temperatures (line 55 page5), but please indicate the present a stratified analysis by three or four categories of indoor temperatures.

When the model was split into indoor temperature tertiles, none of the odds ratios (ORs) were statistically significant, possibly due to small sample size. Because the ORs were small for each degree in temperature, when the range of temperatures was limited there was greater chance of the OR becoming non-significant.

Logistic regression models showing OR of poor health for indoor temperature, stratified by indoor temperature tertiles.

Tertile	Indoor temperature range (°C)	OR	95% CI		P
1 st	7.5-19.7	0.99	0.97	1.02	0.52
2 nd	19.8-21.6	1.04	0.98	1.11	0.17

3 rd	21.7-36.8	1.01	0.98	1.03	0.47
-----------------	-----------	------	------	------	------

We added the word 'tertile' into the Methods section (line 273) and have added the following text into the Results section (lines 337-341)

"When stratified by indoor temperature range, and adjusting for socioeconomic and housing factors, the odds ratios for each tertiles did not differ significantly from one another. In addition, with the smaller sample size and truncated range of temperatures, the relationship between indoor temperature and poor health was no longer significant within any of the tertiles."

Major comment 2: The author should consider the confounding effect of outdoor temperature in the association between indoor temperature and self-rated health. Please conduct multivariable models adjusting for outdoor temperature in Table 3 and Table 4. In addition, stratified analysis by four seasons will be informative for readers. The association between indoor temperature and self-rated health independent of outdoor temperature will suggest valuable information to readers to make a prevention measure.

We included outdoor temperature in the regression models with indoor temperature (Tables 1 and 2) but excluded this from the regression models shown in Table 3 because we assume that indoor temperature lies on the main pathway from outdoor temperature to health (although outdoor temperature may also affect health directly, including by potentially altering outdoor activities and clothing). NB There was no Table 4 in the original manuscript.

We have conducted additional analyses thanks to this reviewer's comment. However, because of the variability by year and region regarding which months are warmer or colder, and how cold they tend to be, we have repeated the analyses stratified by tertile of outdoor temperature, rather than stratifying by season. This is shown in the three sets of additional columns in Table 3.

We have also added the following text to the Discussion (lines 404-407):

Hajat and colleague have shown that 40% of cold-related deaths occur outside the December-March period used for calculating 'excess winter deaths'. [1] For that reason, and given seasonal and temperature differences across the regions of England, we used tertile of outdoor temperature rather than season of other groupings of months.

Major comment 3: We can not find information about the basic characteristics of participants such as age, gender, BMI, household size, Tenure, NS-SEC5, Education, and household income, indoor temperature, and outdoor temperature. Please present the summary data (mean or median of continuous data, and proportion and number of categorical data) of these variables by the seasons of measurement as regular Table. The author states that their characteristics are shown in supplementary Table S1 (page 7, line 12). But I can not find Table S1 in the proof file and uploaded the supplementary file.

We do apologise. We have no idea what happened to this file. We have ensured that it has been uploaded for the revision.

This was created as supplementary material [Table S1] because the journal limits us to a total of five tables and/or figures. For that reason, we have decided to keep this table of characteristics as a supplementary table but we have also included it at the end of the revised manuscript to ensure it is available for the reviewers to see.

We have also added tertile of outdoor temperature to Table S1.

Minor comment 1: If possible, please present the validity of the questionnaire of self-rated health.

This questionnaire has been used each year in the Health Survey for England since 1994. The wording of the question that has been used throughout the Health Survey for England series is

identical to that recommended by: the WHO in its 1996 report on harmonisation of instruments in health interview surveys [2]; the ONS (the Office for National Statistics) in 2008 [3]; and the UK Government Statistical Service harmonisation [3].

We have added this information to the Methods section. We have also expanded our comments on self-rated health in the discussion.

Minor comment 2: Please explain the method to calculate the “% change in OR” in Table 3 in the method section.

We have deleted these from the table so have not added any explanation. We have also amended the text in the preceding paragraph.

Minor comment 3: 3a) Please explain the statistical method of figure 1, and please indicate the parameters of the linear regression, such as regression coefficient, correlation coefficient, and p-value in text.

Figure 1 is the Predicted prevalence of poor health for each degree increase in indoor temperature. It reflects the logistic regression model 1 in Table 3. We have added a footnote to the figure (Lines 449-450):

“Prevalence of poor health predicted by the logistic regression of indoor temperature on poor health, adjusted for age and sex. OR 1.01 (95% CI 1.00-1.02), see table 3, model 1.”

The term “probability” of poor health should be replaced by “Prevalence” (figure 1).

We have changed ‘probability’ to ‘Predicted prevalence’

Minor comment 4: Please add a detailed explanation about the potential mechanism in the discussion section using figure 2.

We have added additional text (lines 361-382), providing a possible explanation for the different results using income or other measures of socio-economic position.

Reviewer: 2

Reviewer Name: Steven M. Schmidt

Institution and Country: Lund University, Sweden

Please state any competing interests or state ‘None declared’: None

In general, I think this is an important topic, and it could be a useful paper.

Thank you

The large sample is rather unique. However, a one time temperature reading may not be very indicative of what the normal indoor temperature is in the areas where the resident spends their time. As most of the previous research in this area has been with older adults, the large number of younger adults could account for some of the findings. Younger adults, for one, spend much less time at home than older adults, so the indoor temperature at home has a smaller impact on their health compared to older adults.

As the reviewer notes, one of the strengths of this study is its size. We comment on the inadequacy of a single temperature measurement in the limitations section of the discussion. We rephrased that sentence to emphasise this more clearly (lines 397-400):

“The other main limitation was the measurement of indoor temperature at a single time point for each participant in whichever room they were ‘hosting’ the survey nurse, generally in the living room or kitchen.”

There are a number of issues addressed below that would need to be resolved as some parts of the paper are very difficult to understand in the present state, so it is difficult to fully judge the findings at this time.

The main focus seems to be on cold homes in the abstract objective, but you do assess a full range of temperatures in at least one analysis. The focus should be more clear in the background as well.

Although extreme cold is the major focus of this paper, overheating in the summer also has adverse health consequences. Cold is the primary focus because that is the predominant exposure for temperature-related health effects.

We have amended the abstract, adding a sentence to the Objective (lines 43-44):

“Overheating in the summer also has adverse health consequences.”

And added this sentence to the penultimate paragraph in the Introduction (lines 125-128):

“There are also deaths in England due to excessive temperatures during heatwaves,[4] so both high and low indoor temperatures warrant studying in relation to associations with health, although cold is the primary focus because that is the dominant exposure in England.”

Please be consistent with terminology throughout. For example, in the abstract you talk about ill-health, but then use poor health in the body of the paper. You also talk about “ill households” in the abstract but a household cannot be ill, only a person can. In general, the abstract uses different language and terms compared to the body of the paper.

Thank you for pointing this out. We have changed the text to use poor health (line 60) and changed ‘ill households’ to ‘people with worse health’ (line 68).

Research Hypotheses: The second hypothesis is “poor indoor temperature control would be associated with worse general health.” However, you did not assess temperature control. It is not clear what you mean by poor indoor temperature control?

Thank you for this comment. We were assuming that people would control the indoor temperature inasmuch as they were able to, to maintain a home that was not unduly cold (or hot). We have rephrased this (line 132) as ‘low indoor temperature’.

Methods: I would like to see a description of the study design at the start of the methods. In the title and abstract you state that it is a “cross-sectional study” but this is not stated in the methods or elsewhere in the body of the paper. A little more detail of the design would also be nice.

We have added a new section ‘Study design’ at the start of the Methods section, where we moved some of the existing text, and provided a few additional details and references:

“Study design

This study is based on the Health Survey for England (HSE), an annual, cross-sectional, household, health examination survey which has run since 1991. A new, nationally-representative sample of the free-living general population is selected each year, using a stratified, two-stage selection process. The sampling frame is the small addresses Postal Address File. The survey varied in size each year from around 9,000 to 16,000 households in the study sample. Data were collected first by an interviewer, who also measured height and weight; a nurse visit followed for participants who agreed. The nurse asked further questions and took biophysical measurements, including blood pressure. More details are reported elsewhere,[5,6] including the Methods report for the annual report published each year (e.g.[7]).

For this survey, we used data from each year HSE 2003 to HSE 2014 excluding 2004, when the core general population sample did not have a nurse visit. These years were selected because identical variables have been coded consistently from 2003 onwards; 2014 was the most recent dataset available in the UK Data Archive at the time the analyses commenced.

Participants: The Participants section has a lot of detail that is not per se related to the description of participants. Most of the first paragraph in the participants section is used to describe the HSE survey. I would suggest splitting this into two separate sections with one describing your actual sample and the other describing the survey and data collection procedures.

We have done this.

When the survey is described it would also be good to give a short description of how the data were combined. Were all people from all years combined into a single data set? Was any adjustment made for the year of assessment? Over an 11 year period there could be other factors that could have influenced perceived health.

We have added the following sentence to the end of the first paragraph of the 'Statistical analysis' section (lines 249-250):

"The data were combined into a single dataset."

No adjustment was made for survey year. Trend table data show that from 2003 to 2014 between 75% and 76% of adults (76%-78% of men and 74%-76% of women) reported very good or good health, with the percentages fluctuating within this narrow band year-to-year (<https://digital.nhs.uk/data-and-information/publications/statistical/health-survey-for-england/2018/health-survey-for-england-2018-data-tables>). Thus the percentage with poor health, the combined other three response categories, varied very little for all adults - between 24% and 25%.

Data: The second paragraph is hard to follow and seems to have some incorrect use of punctuation. Please check for proper use of colon and semi colon in first sentence.

We have used semicolons to separate items in a list, as the individual items are long or contain internal punctuation. We deleted the colon in the first sentence as the reviewer dislikes it. However, this comment may now be redundant as we have expanded the section on data collection and categorisation, as requested below.

I would recommend re-writing this second paragraph of Data to clarify what was assessed and what the response choices were for each item. I had to look down to table 1 to really start to understand what the variables were. This should already be clearly explained in the methods. The response choice need clear explanations for an international readership to understand. It seems very UK specific, so I have trouble knowing what they mean unless defined.

We have added subheadings for the different types of variables, as well as adding more detail. For example (lines 199-206; 210-218; 219-223):

"Ill health

"The other self-reported health data used in these analyses included the presence of limiting or non-limiting longstanding illness; a respiratory condition; or a cardiovascular condition. Participants were first asked if they had any longstanding illness or condition. Those who responded positively were then asked if it limited their daily activities, and asked what the condition(s) were, to a maximum of six. These were then coded into ICD-10 chapters, from which responses from participants volunteering a respiratory condition (chapter J) and/or a cardiovascular condition (chapter I) were identified."

and

“Socio-economic data

“Three variables were used for socio-economic position. Education level was categorised as degree or equivalent, qualification below degree, or no qualification. Equivalised household income, which adjusts for the numbers of adults and of children in the household, was divided into quintiles. The National Statistics-Socio-Economic Classification of occupation in five categories (NS-SEC5)³¹ was derived from detailed information about employment status, occupation, size of organisation, and managerial responsibility. The categories were managerial & professional, intermediate, small employers, lower supervisory & technical, and semi-routine.”

and

“Housing

Housing information comprised the number of occupants in the household, and housing tenure. The latter was categorised as owner-occupied (e.g. owned outright or with a mortgage), rented privately, or socially rented (i.e. from a housing association or local government).”

For example in the three categories for education, what type of degree (university, primary school, secondary school)? What is and “other qualification”?

In the UK, ‘degree’ signifies university education. The international recommendation for classifying educational achievement to be as comparable as possible across countries is to use ‘no qualification’, ‘degree or equivalent’, and ‘any qualification below degree standard (which we called ‘other qualification’).

We have amended ‘other qualification’ to ‘qualification below degree’ (line 212), which was already used in the tables.

Statistical analysis: I am not sure I understand your justification for categorizing age. Please elaborate to explain this.

Using age-groups avoids pre-specifying the shape of any relationship with age. It is a common practice where the exact shape of the relationship with age is not the primary research question.

If you are going to include a category of participants as “other or missing”, then I think they should also be shown in the results.

It is a common practice where there is a substantial proportion of cases where data on potential confounders are missing to include these as a separate category when conducting the analyses to avoid excluding other data from participants which not only reduces the sample size but could introduce bias. We have not shown the results for such ‘missing’ categories as they do not provide useful information in themselves: they are an unknown (on that measure) and possibly heterogenous group.

The analyses are not described for the results presented in Table 1. Please add it. It seems that the temperature was dichotomize into <18 degrees and 18 degrees or higher, but this is not stated.

Thank you for pointing out these omissions. We have revised the text (lines 260-265) as follows:

“Associations between socioeconomic and housing variables and indoor temperature were explored in two ways. Logistic regression models were created with indoor temperature <18°C as the outcome variable (compared with indoor temperature of 18°C or above) as well as linear regression models with indoor temperature as a continuous variable. For both sets of models, socioeconomic and housing variables were included as explanatory variables.”

In general it is hard to match what is described in this section with the results. Please elaborate and clarify what was done.

We apologise for this and hope the revised manuscript is clearer.

Results

There is no "Supplementary Table S1" that I could find. There needs to be a very good description of the study sample, but I cannot judge this at this point as it seems like it might be in the missing table.

We are sorry that Table S1 did not appear in the reviewers' copy of the manuscript. We have ensured that it has been uploaded and, in addition, have pasted it at the end of the revised manuscript.

In the first paragraph of results indoor temperature is presented but outdoor temperature is not. Please show the range and mean for outdoor temperature as well.

We have done this (lines 289-290):

"The outdoor temperature ranged from -1.2°C to 20.5°C, with a mean of 10.1°C (SD 4.6)."

Please note that outdoor temperatures were not measured for each participant but were obtained from the publicly available Met Office data: we used the monthly mean outdoor temperature for the region in which the HSE participant lived.

Please organize the paper so that the methods (Statistical analyses) and results are the same order as the hypotheses. It is currently not consistent

We hope that the re-organised and expanded Methods section now fits the order in which we present the (expanded) results.

There seems to be an extra figure with no explanation.

We provided a graphical abstract, which many journals are now using. It is the same as Figure 2.

The last row of Table 2 has some numbers in bold, but I am not sure why.

Thank you for pointing this out. We have corrected this.

In the section "Indoor temperature and health" you describe the increase in odds, but then you refer to figure 1 where you label the axis probabilities. What is really being presented in the two parts? Probabilities are not mentioned in any of the text and only in the figure. Several terms are used in the results: odds, odds ratios, and probability, but it is not clear if you have calculated different things or if these terms are, incorrectly, being used interchangeably.

Thank you for pointing this out. The logistic regression results in Table 3 were reported as odds ratios. Figure 1 uses those odds ratios to predict the prevalence of poor health. We have amended the title and the y axis of Figure 1, and made the results consistent.

Discussion: I would like to see more discussion about what the finding mean. You simply state that "the relationship between socioeconomic and housing factors with indoor temperature was complicated", but you do not really make any effort to try to explain that complexity.

We have added additional text (lines 361-382), providing a possible explanation for the different results using income or other measures of socio-economic position.

Besides mortality, a number of studies have looked at the impact of temperature on blood pressure, and have shown larger increases in blood pressure with lower temperatures and higher ages. Perhaps the indoor temperature does not have much of an association among younger ages, and you may find a different relationship at higher ages. As there was a nurse visit what types of health outcomes were assessed that you could use as an alternative to self-reported health? I would say blood pressure for one.

It is because of the well-known relationship between temperature and blood pressure that the survey nurses measure the indoor temperature prior to measuring blood pressure, and thus we have this set of indoor temperatures from a large number of individuals. However, this manuscript is not examining associations between BP and temperature, which has been done both using Health Survey for England data and elsewhere.

This study was designed to look at self-reported general health. Thus the existence of other measures is not relevant to our choice of outcome.

We have, however, expanded the discussion to include additional statements and references to the associations between temperature and aspects of health (lines 411-416):

“While the subjectivity of self-rated health makes it vulnerable to information bias, it has been shown to be highly predictive of future mortality and health outcomes, and is used frequently across the scientific literature.[8,9] Self-rated health is strongly associated with total mortality [10–14] and with some specific causes of death [15,16] but not with external causes.[15]. It also predicts functional ability.[17]. Self-rated health is explained more by measures of mental and physical health than by demographic or socio-economic factors.[18] Apart from the well-known association between temperature and blood pressure,[19–22] there is less evidence pointing at more specific health outcomes related to temperature. Thus, as this is the first study of its kind, the broadness of general health as the outcome was a strength rather than a weakness. The differences between the entire sample and those with complete data on self-rated health and temperature was primarily due to non-response to the nurse visit among interviewees. This was dealt with by use of non-response weighting.[6]”

References:

- 1 Hajat S, Gasparrini A. The Excess Winter Deaths Measure: Why Its Use Is Misleading for Public Health Understanding of Cold-related Health Impacts. *Epidemiology* 2016;**27**:486–91. doi:10.1097/EDE.0000000000000479
- 2 de Bruin A, Picavet HSJ, Nossikov A, editors. *Health interview surveys: towards international harmonization of methods and instruments*. Copenhagen: : World Health Organization, Regional Office for Europe 1996. https://www.euro.who.int/__data/assets/pdf_file/0017/111149/E72841.pdf?ua=1 (accessed 14 Jul 2020).
- 3 Nixon O. General health harmonised principle. Gov. Stat. Serv. 2020. <https://gss.civilservice.gov.uk/policy-store/general-health/> (accessed 14 Jul 2020).
- 4 Public Health England. PHE heatwave mortality monitoring, Summer 2019. London: : PHE https://assets.publishing.service.gov.uk/government/uploads/system/uploads/attachment_data/file/841320/PHE_heatwave_report_2019.pdf (accessed 24 Jul 2020).
- 5 Mindell J, Biddulph JP, Hirani V, *et al*. Cohort Profile: The Health Survey for England. *Int J Epidemiol* 2012;**41**:1585–93. doi:10.1093/ije/dyr199
- 6 Mindell JS, Goesswald A, Kamtsiuris P, *et al*. Sample selection, recruitment and participation rates in health examination surveys in Europe – experience from seven national surveys. *BMC Med Res Methodol* 2015;**15**:78. doi:10.1186/s12874-015-0072-4
- 7 Craig R, Fuller E, Mindell J (Eds.). Health Survey for England 2014. Volume 2. Methods and documentation. Leeds: : NHS Digital 2015. <https://digital.nhs.uk/data-and-information/publications/statistical/health-survey-for-england/health-survey-for-england-2014>

- 8 Simon JG, De Boer JB, Joung IMA, *et al.* How is your health in general? A qualitative study on self-assessed health. *Eur J Public Health* 2005;**15**:200–8. doi:10.1093/eurpub/cki102
- 9 Jylhä M. What is self-rated health and why does it predict mortality? Towards a unified conceptual model. *Soc Sci Med* 2009;**69**:307–16. doi:10.1016/j.socscimed.2009.05.013
- 10 Heistaro S, Jousilahti P, Lahelma E, *et al.* Self rated health and mortality: a long term prospective study in eastern Finland. *J Epidemiol Community Health* 2001;**55**:227–32. doi:10.1136/jech.55.4.227
- 11 Idler EL, Benyamini Y. Self-rated health and mortality: a review of twenty-seven community studies. *J Health Soc Behav* 1997;**38**:21–37.
- 12 Kaplan GA, Camacho T. Perceived health and mortality: A nine-year follow-up of the Human Population Laboratory Cohort. *Am J Epidemiol* 1983;**117**:292–304. doi:10.1093/oxfordjournals.aje.a113541
- 13 Møller L, Kristensen TS, Hollnagel H. Self rated health as a predictor of coronary heart disease in Copenhagen, Denmark. *J Epidemiol Community Health* 1996;**50**:423–8. doi:10.1136/jech.50.4.423
- 14 Mossey JM, Shapiro E. Self-rated health: a predictor of mortality among the elderly. *Am J Public Health* 1982;**72**:800–8. doi:10.2105/AJPH.72.8.800
- 15 Benjamins MR, Hummer RA, Eberstein IW, *et al.* Self-reported health and adult mortality risk: An analysis of cause-specific mortality. *Soc Sci Med* 2004;**59**:1297–306. doi:10.1016/j.socscimed.2003.01.001
- 16 Wannamethee G, Shaper AG. Self-assessment of Health Status and Mortality in Middle-Aged British Men. *Int J Epidemiol* 1991;**20**:239–45. doi:10.1093/ije/20.1.239
- 17 Idler EL, Kasl SV. Self-Ratings of Health: Do they also Predict change in Functional Ability? *J Gerontol B Psychol Sci Soc Sci* 1995;**50B**:S344–53. doi:10.1093/geronb/50B.6.S344
- 18 Singh-Manoux A. What does self rated health measure? Results from the British Whitehall II and French Gazel cohort studies. *J Epidemiol Community Health* 2006;**60**:364–72. doi:10.1136/jech.2005.039883
- 19 Umishio W, Ikaga T, Kario K, *et al.* Cross-Sectional Analysis of the Relationship Between Home Blood Pressure and Indoor Temperature in Winter: A Nationwide Smart Wellness Housing Survey in Japan. *Hypertension* 2019;**74**:756–66. doi:10.1161/HYPERTENSIONAHA.119.12914
- 20 Wang Q, Li C, Guo Y, *et al.* Environmental ambient temperature and blood pressure in adults: A systematic review and meta-analysis. *Sci Total Environ* 2017;**575**:276–86. doi:10.1016/j.scitotenv.2016.10.019
- 21 Zhao H, Jivraj S, Moody A. “My blood pressure is low today, do you have the heating on?” The association between indoor temperature and blood pressure. *J Hypertens* 2019;**37**:504–12. doi:10.1097/HJH.0000000000001924
- 22 Cabrera SE, Mindell JS, Toledo M, *et al.* Associations of Blood Pressure With Geographical Latitude, Solar Radiation, and Ambient Temperature: Results From the Chilean Health Survey, 2009–2010: Table 1. *Am J Epidemiol* 2016;**183**:1071–3. doi:10.1093/aje/kww037

VERSION 2 – REVIEW

REVIEWER	Keigo Saeki Department of Epidemiology, Nara Medical University School of Medicine, Japan
REVIEW RETURNED	28-Aug-2020
GENERAL COMMENTS	General Comment Unfortunately, even after the revision, the quality of the manuscript did not improve sufficiently in the statistical analysis, interpretation of the results, and discussion about the mechanism. Major comment 1

	The author added the stratified analysis by tertile groups and did not find a significant association between indoor temperature and self-reported health. The author should consider the reason why the significant association disappeared after stratification other than the smaller sample size because the total sample size of the present study is 74736. Major comment 2 The author conducted an additional stratified analysis by tertile groups of outdoor temperatures (Table 3). However, the description of the odds ratio by outdoor temperature and the interpretation of the result is insufficient. Major comment 3 The basic characteristics of the participants should be presented as a regular table.
--	--

REVIEWER	Steven Schmidt Lund University Sweden
REVIEW RETURNED	10-Sep-2020

GENERAL COMMENTS	Thank you for working to address the issues raised during the previous review. I think the paper has much better now and easier to read and understand. Most of my previous points were adequately addressed. I have a few remaining minor points below from before and a few new minor points now that table S1 is available. Lines 171-173 state, “However, among the categorical variables, only respiratory disease had non-significant differences in their distribution within the two samples.” The wording is awkward. If it is not significant then there cannot be a difference. It would be clearer and more direct to write something like this: However, all categorical variables, except for respiratory disease, had significant differences between the analytic sample and the excluded participants. Line 199, Thank you for improving the consistency of terminology throughout. It is much clearer now. However the heading on this line is still written “Ill Health”, which should be changed to poor health based on your decision to use this term throughout the paper. Lines 245-250, Thank you for clarifying the other/missing group. Related to my previous review comment, “If you are going to include a category of participants as “other or missing”, then I think they should also be shown in the results,” I agree that it is quite typical practice to include people in such a category in order to keep your sample size. However, I disagree with your response that showing the result for these people “do not provide useful information.” I believe that if you have something in the analysis it should be reported. You can state in your discussion that you think this is not useful information, but by having it in the results it shows the entire picture of your sample and allows the reader to judge the complete analysis for themselves. Table S1, Could you please add at the top of the table, row 2, next to the name of each sample and excluded group the total N for each of these 3 groups?
--

	Line 292, I am curious. Is there a recommended maximum indoor temperature in the UK. 36.8 is pretty hot. Starting at line 296 in the logistic regressions. You point out the specific significant OR, but they are hard to interpret unless you know which category was the reference. Please add this to the text. I can see it in the table, but I don't what to have to keep jumping back and forth between the table and the text to try to understand the findings. Starting at line 397 related to limitation of 1 temperature measurement. It is good that you acknowledge this limitation, but please provide some explanation of how this could impact the interpretation of the results as well. It could also be good to add something in the limitations about missing data particularly related to income as 16% of the sample was missing on this.
--	---

VERSION 2 – AUTHOR RESPONSE

Reviewer: 1

Reviewer Name: Keigo Saeki

Institution and Country: Department of Epidemiology, Nara Medical University School of Medicine, Japan

Please state any competing interests or state 'None declared': None declared

General Comment

Unfortunately, even after the revision, the quality of the manuscript did not improve sufficiently in the statistical analysis, interpretation of the results, and discussion about the mechanism.

Major comment 1

The author added the stratified analysis by tertile groups and did not find a significant association between indoor temperature and self-reported health. The author should consider the reason why the significant association disappeared after stratification other than the smaller sample size because the total sample size of the present study is 74736.

We reanalysed the data stratified by tertile of indoor temperature adjusted for housing, socio-economic, and demographic factors (see Sensitivity analysis table below). We have amended the text in the results as follows (lines 323-329):

“When stratified by indoor temperature range, and adjusting for socioeconomic and housing factors, the odds ratios for each indoor temperature tertile did not differ significantly from one another (sensitivity analysis, data not shown). ~~In addition, w_~~With the smaller sample size and truncated range of temperatures, the relationship between indoor temperature and poor health was no longer significant within ~~any of the~~ upper two tertiles. This may be due to truncation in the distribution of indoor temperature, meaning that only the lowest tertile included the low temperatures associated with poor health.”

Sensitivity analysis: Logistic regression models showing adjusted OR of poor health for indoor temperature, stratified by indoor temperature tertiles

Tertile	Temperature range	OR	SE	95% CI	P
1st tertile	7.5-19.7	1.028	0.0140	1.001 1.056	0.043

2nd tertile	19.8-21.6	1.026	0.0326	0.964	1.092	0.420
3rd tertile	21.7-36.8	0.996	0.0133	0.970	1.023	0.772
All tertiles	7.5-36.8	1.017	0.0048	1.007	1.026	<0.001

Adjusted for hh size, nssec, income, qualifications, and age and sex

Major comment 2

The author conducted an additional stratified analysis by tertile groups of outdoor temperatures (Table 3). However, the description of the odds ratio by outdoor temperature and the interpretation of the result is insufficient.

We are confused by this comment as it is general practice for journals not to want details presented in tables to be reiterated in the text. We have mentioned Table 3 (now Table 4) in this paragraph and have amended the caption for this table to:

Table 3-4 - Logistic regression models showing odds ratios of poor health for indoor temperature, stratified by tertile of outdoor temperature

We have also amended the text to provide some more details (lines 335-345):

When regression models were repeated stratified for tertile of outdoor temperature, we found the ORs of indoor temperature on poor health were stronger for the coldest tertile (i.e. <7°C). For example, the adjusted Odds Ratios for Models 7 and 9, which included adjustment for housing variables, were 1.04 (1.02-1.05) in the lowest tertile but were not significantly different from 1.0 for the middle and highest tertile of outdoor temperature (Table 4~~3~~). This may be because the lower the outdoor temperature, the more likely it is that the occupant will choose to turn on their heating and therefore choose what temperature their home is at, which will depend on the characteristics of both the home and the residents. On warmer days, occupants are less likely to turn on their heating, and therefore the indoor temperature will be determined more by the outdoor temperature than the characteristics of the occupant, and as such will be less associated with any characteristics of the occupant. ~~The point estimates for the warmer tertiles of outdoor temperature were similar to the overall ORs but were not significantly above unity.~~

Major comment 3

The basic characteristics of the participants should be presented as a regular table.

We have added a new Table 1, describing the basic characteristics of the participants included in the analysis, as well as those excluded as they were incomplete on the vital variables.

Reviewer: 2

Reviewer Name: Steven Schmidt

Institution and Country: Lund University, Sweden

Please state any competing interests or state 'None declared': None declared

Thank you for working to address the issues raised during the previous review. I think the paper has much better now and easier to read and understand. Most of my previous points were adequately addressed.

Thank you.

I have a few remaining minor points below from before and a few new minor points, now that table S1 is available.

Lines 171-173 state, “However, among the categorical variables, only respiratory disease had non-significant differences in their distribution within the two samples.” The wording is awkward. If it is not significant then there cannot be a difference. It would be clearer and more direct to write something like this:

However, all categorical variables, except for respiratory disease, had significant differences between the analytic sample and the excluded participants.

Thank you for this suggestion, which we have incorporated (lines 159-161).

Line 199, Thank you for improving the consistency of terminology throughout. It is much clearer now. However the heading on this line is still written “Ill Health”, which should be changed to poor health based on your decision to use this term throughout the paper.

Thank you for spotting this. We have made the change (line 187).

Lines 245-250, Thank you for clarifying the other/missing group. Related to my previous review comment, “If you are going to include a category of participants as “other or missing”, then I think they should also be shown in the results,” I agree that it is quite typical practice to include people in such a category in order to keep your sample size. However, I disagree with your response that showing the result for these people “do not provide useful information.” I believe that if you have something in the analysis is should be reported. You can state in your discussion that you think this is not useful information, but by having it in the results it shows the entire picture of your sample and allows the reader to judge the complete analysis for themselves.

Although we disagree, we have added these data into Tables 2 and 3 (as renumbered, allowing for the new Table 1)

Table S1, Could you please add at the top of the table, row 2, next to the name of each sample and excluded group the total N for each of these 3 groups?

Thank you for this suggestion, which we have incorporated.

Table S1. Characteristics of participants

Variable	Category	Analytical sample ^a (N=74,736)		Excluded from analysis ^b (N=2065)		Study sample ^c (N=76,801)	
		N*	%	N	%	N	%
Gender ^d	Men	33,526	45%	533	26%	34,059	44%
	Women	41,210	55%	1,532	74%	42,742	56%

Line 292, I am curious. Is there a recommended maximum indoor temperature in the UK. 36.8 is pretty hot.

No there is no recommended maximum temperature. This is reflected in the WHO guidance, which gives a recommended minimum temperature but not a maximum one. Similarly, there is a legal minimum temperature for workplaces in the UK but no maximum allowable temperature.

Starting at line 296 in the logistic regressions. You point out the specific significant OR, but they are

hard to interpret unless you know which category was the reference. Please add this to the text. I can see it in the table, but I don't what to have to keep jumping back and forth between the table and the text to try to understand the findings.

We have amended the text as follows (lines 275-287):

Households with more than one occupant and households with higher equivalised household incomes were associated with ~~had~~ a lower odds of an indoor temperature <18°C (Table 42) and higher mean indoor temperatures (Tables 23 and S1) than single person households, as did households with higher equivalised household incomes compared with those with lower incomes. The adjusted OR (Table 42) and β -coefficient (Table 23) for both household size and income showed a gradient across all categories. Higher educational attainment, and private sector rented houses (compared with owner occupied housing) were associated with lower mean indoor temperatures (Table 23) and a higher odds of the indoor temperature being <18°C (Table 42). NS-SEC5 was variably associated with low temperature; compared with those in managerial and professional occupations, we found ~~(reduced odds-adjusted OR among those in intermediate occupations but higher odds among small employers),~~ and with mean indoor temperature (positive association in lower supervisory and technical households but inverse association among small employers) in the fully adjusted models, ~~compared with those in managerial and professional occupations.~~

Starting at line 397 related to limitation of 1 temperature measurement. It is good that you acknowledge this limitation, but please provide some explanation of how this could impact the interpretation of the results as well.

We have changed the text (lines 403-410):

The third main limitation was the measurement of indoor temperature at a single time point for each participant in whichever room they were 'hosting' the survey nurse, generally in the living room or kitchen. Huebner and colleagues have shown four different profiles in living room temperature variability across the day, varying from bimodal (morning and evening peaks), steadily increasing (shallow and steep) to flat with little variation.[38] Using the single time point temperature reading in the analysis requires an assumption that the snapshot temperature reading for each participant reflects an average temperature for the home over a significant period of time.

It could also be good to add something in the limitations about missing data particularly related to income as 16% of the sample was missing on this.

We have added the following text (lines 395-400):

In addition, there was some missing data among the analytic sample (Table 1). Income is the variable with the highest level of missingness. It is likely that this group contains disproportionate numbers of households both of high and of low income. Tables 2 and 3 show that this group have a similar odds of low temperature as the 2nd lowest quintile, and a similar coefficient for linear temperature as the 3rd quintile in the fully adjusted models.